# Microalgae Production on Biogas Digestate in Sub-Alpine Region of Europe—Development of Simple Management Decision Support Tool

**Lara Resman [1], Maja Berden Zrimec [2], Vid Žitko [1], Borut Lazar [2], Robert Reinhardt [2], Ana Cerar [2] and Rok Mihelič [1,*]**

[1] Department of Agronomy, Biotechnical Faculty, University of Ljubljana, Jamnikarjeva 101, 1000 Ljubljana, Slovenia; lara.resman@bf.uni-lj.si (L.R.); vid.zitko@bf.uni-lj.si (V.Ž.)

[2] AlgEn, Algal Technology Centre, LLC, Brnčičeva 29, 1231 Ljubljana, Slovenia; maja@algen.si (M.B.Z.); borut@algen.si (B.L.); robert@algen.si (R.R.); ana@algen.si (A.C.)

\* Correspondence: rok.mihelic@bf.uni-lj.si; Tel.: +386-1-3203-201

**Abstract:** In a one-and-a-half-year study conducted in the ALS6 region in Europe (Ljubljana, Slovenia), the cultivation of microalgae in anaerobic digestate from food waste, mainly *Scenedesmus dimorphus* and *Scenedesmus quadricauda*, was investigated in three ponds (1260 L each) under a greenhouse. The effects of changing digestate quality and quantity as well as seasonal fluctuations on the productivity of the microalgae were investigated in three stages: Learning/Design (SI), Testing (SII), and Verification/Calibration (SIII). A decision support tool (DST) was developed using easy-to-measure parameters such as pH, temperature, electrical conductivity, mineral nitrogen forms and physical, biological parameters (OD, delayed fluorescence intensity). To control optimal pond operation, we proposed the photosynthetic culture index (PCI) as an early indicator for necessary interventions. Flocculation and nitrite levels (above 3 mg $NO_2$-N $L^{-1}$) were signals for the immediate remediation of the algae culture. Under optimal conditions in summer SIII, an average algal biomass production of $11 \pm 1.5$ g $m^{-2}$ day$^{-1}$ and a nitrogen use efficiency of $28 \pm 2.6$ g biomass/g N-input were achieved with the developed DST. The developed DST tool was, in this study, successfully implemented and used for the cultivation of microalgae consortia predominated by *Scenedesmus dimorphus* and *S. quadricauda* with biogas digestate. DST offers the possibility to be modified according to producers' specific needs, facility, digestate and climate conditions, and as such, could be used for different microalgae cultivation processes with biogas digestate as a food source.

**Keywords:** circular economy; anaerobic digestion; food waste digestate; microalgae *Scenedesmus*; open ponds; decision support tool

## 1. Introduction

The large-scale cultivation of algal biomass in wastewater streams is an attractive technology for putting circular bioeconomy theory into practice. Algae extract and recycle nutrients, organic carbon, and minerals that would otherwise be lost to the environment [1,2]. They also sequester $CO_2$ from the atmosphere. In wastewater treatment plants, they reduce energy costs by 50% by providing bacteria with the oxygen needed for biological treatment [3]. In turn, wastewater enables more economical large-scale production without the need for large quantities of quality water and expensive commercial growth media [4]. Large-scale algal cultivation offers an interesting opportunity for multiple industries and agriculture, as algae can grow in a variety of waste substrates, and the biomass produced can be utilized for a variety of products, providing additional financial streams, instead of costs, for wastewater disposal or conventional treatment [5,6].

Recently, microalgae cultivated in wastewater have been well utilized in agricultural products. Organic fertilizers from microalgae are available in the market, and their benefi-

cial effects on soil and plants have been demonstrated on many occasions [7,8]. Research on microalgae shows promising results when using microalgae derived products in agriculture: microalgal biofertilizers improve soil health [9,10]; microalgal biostimulants have positive effects on development, growth, and yields of crops [11]; and microalgae may be considered potential biocontrol agents—biopesticides [8].

While digestate can be used directly as an organic fertilizer due to its high nutrient content, it presents logistical and storage challenges [12,13] associated with high greenhouse gas emissions [14]. In addition, the composition of digestate can vary widely, making its application in fields a challenge for farmers [15,16]. Algae cultivated in digestate can help stabilize it by incorporating nutrients into their biomass. The stabilization allows for easier storage, transport, and application, reducing the environmental footprint [17]. Green microalgae have been widely researched for the treatment of liquid digestate and are efficient in removing nutrients and organic contaminants from this source [18]. Microalgae have the capacity to utilize dissolved carbon dioxide and nutrients like ammonia, nitrates, nitrites, and phosphates. This characteristic renders them highly effective in treating liquid digestate; for example, the potential to reduce nitrogen, phosphates, and total COD by up to 70%, 57%, and 95%, respectively, was demonstrated with microalgae consortia [19], and even higher efficiencies are regularly reported [20,21].

Open algal ponds are a low-cost technology, and their maintenance is relatively simple [18]. They can be integrated into the existing technology system as side-streams without major restructuring, which makes them appealing to industrial operators. In the case of biogas plants, ponds can be installed on-site to treat the digestate, which significantly reduces the costs and environmental impact [19]. Nevertheless, in the sub-alpine climate, the operation of the algae in pond culture is hampered by the low temperature, fog, and low solar irradiation in the colder months, typically resulting in the low conversion of digestate nutrients to algal biomass, which is not attractive to biogas plant operators.

The main obstacles in scaling up biomass production in the conventional system are high processing costs and low efficiency [22,23]; therefore, the use of digital technology could improve productivity effectively and efficiently manage microalgal biomass production [24]. The monitoring and control of parameters such as pH, T, and nutrient status is crucial in microalgae culture because it not only increases the production and quality of microalgae biomass but also prevents critical conditions for the culture that may jeopardize cultivation [25]. Easy-to-use monitoring and control, coupled with a decision support tool for microalgal cultivation, could be of great assistance to microalgal facility operators [26].

This work aimed to establish and test the algal production on the liquid part of biogas digestate in the Alpine South climate, ALS6, and to develop and test a simple decision support tool (DST) for microalgae cultivation in liquid anaerobic digestate. During different stages of the year-and-a-half experiment, the influence of different digestate concentrations in various seasons on the microalgae growth and digestate utilization efficiency were studied in raceway ponds inside a simple, unheated, foil-covered greenhouse.

## 2. Materials and Methods

### 2.1. Microalgae Cultivation

Microalgae were cultivated in three identical raceway ponds situated in a simple double-foil tunnel greenhouse in Ljubljana, Slovenia. A one-and-a-half-year study in the ALS6 environmental zone, characterized by a temperate climate with Mediterranean and continental influences [27] (Figure 1), included all four seasons characteristic of the region.

The greenhouse, measuring 20 m × 9 m, was equipped with double-layer PE foil, insect net protection and a temperature control system. During winter, the greenhouse was heated only to prevent freezing. The open raceway ponds were made of ABS plastic and coated with PMMA material. Each pond had a capacity of 1260 L (6.3 m$^2$ area, depth of culture 20 cm) and was equipped with a paddlewheel, a $CO_2$ diffuser, and a sensor system for pH regulation (Figure 2).

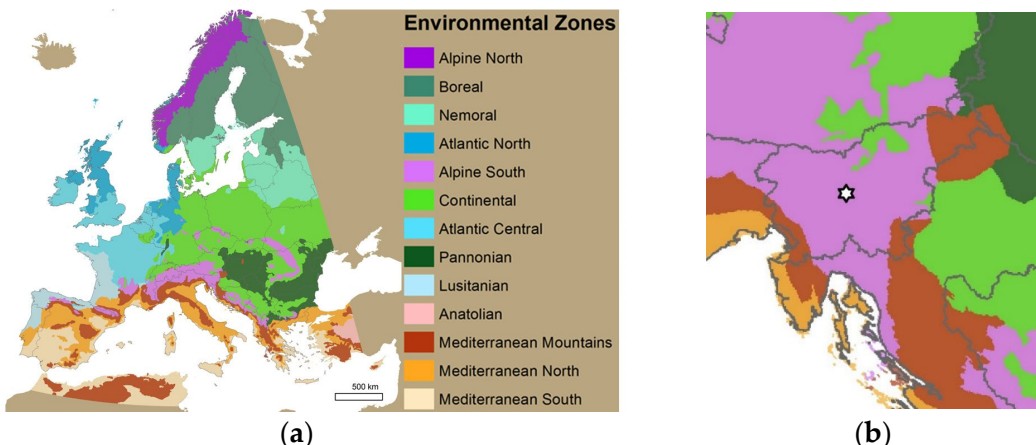

(**a**)　　　　　　　　　　　　　　　　(**b**)

**Figure 1.** Climatic stratification of the environment of Europe: (**a**) All environmental zones of Europe; (**b**) Environmental zones in Slovenia, Ljubljana (white star), is located in the Alpine South. Pictures: modified dataset Metzger, Marc J. (2018). The Environmental Stratification of Europe, [dataset]. University of Edinburgh. https://doi.org/10.7488/ds/2356 [28].

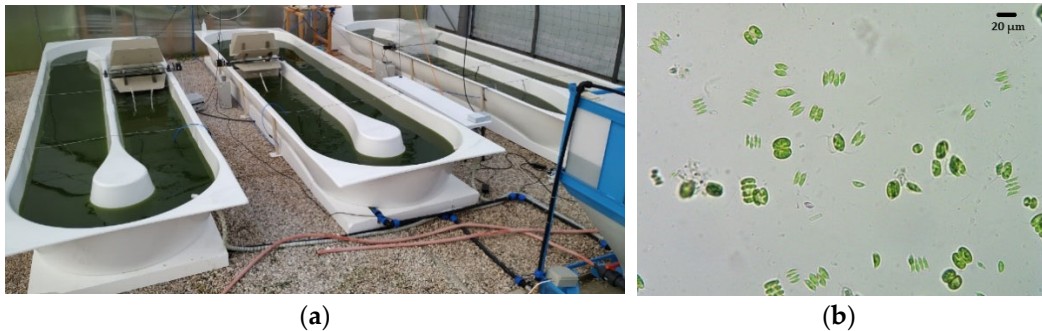

(**a**)　　　　　　　　　　　　　　　　(**b**)

**Figure 2.** Production of microalgae. (**a**) Three raceway ponds installed in foil greenhouse; (**b**) Inoculum for three raceway ponds with predominant species *Scenedesmus dimorphus* and *Scenedesmus quadricauda* (under light microscope (AmScope T690C-PL-5M, Irvine, CA, USA), 400× magnification).

*2.2. Microalgae Inoculum*

The microalgae inoculum was sourced from a greenhouse pilot plant with a raceway pond at the food-waste biogas plant (KOTO, Ljubljana, Slovenia), where microalgae culture has been growing in digestate since 2015 [29]. The prevailing microalgae species in the inoculum culture identified were *Scenedesmus dimorphus* (Turpin) Kützing 1834 [30] and *S. quadricauda* (Turpin) Brébisson 1835 [31] (Figure 2).

*2.3. Digestate Usage*

The liquid fraction of anaerobic digestate from food waste was delivered in batches throughout the year. Based on previous experience, pretreatment of the digestate was not required. The chemical characteristics of the digestate batches are detailed in Table 1.

**Table 1.** Digestate characterization.

| Parameter | Unit | Digestate Batches | | | | |
| | | D1 | D2 | D3 | D4 | D5 |
|---|---|---|---|---|---|---|
| pH | - | 7.75 ± 0.15 | 8.02 ± 0.26 | 7.75 ± 0.10 | 8.16 ± 0.52 | 8.24 ± 0.09 |
| EC | $\mu S\ cm^{-1}$ | 13,026 ± 355 | 13,118 ± 300 | 11,995 ± 204 | 8770 ± 1432 | 11,310 ± 289 |
| $NO_3$-N | $g\ kg^{-1}\ d.m.^{-1}$ | 0.33 ± 0.03 | 0.08 ± 0.04 | 0.06 ± 0.05 | 0.03 ± 0.02 | 0.002 ± 0.001 |
| $NH_4$-N | $g\ kg^{-1}\ d.m.^{-1}$ | 140 ± 2 | 571 ± 45 | 119 ± 28 | 15 ± 5 | 68 ± 4 |
| TS | $g\ L^{-1}$ | 8.13 ± 0.12 | 4.98 ± 0.34 | 5.08 ± 0.80 | 5.88 ± 0.60 | 4.02 ± 0.42 |
| OM | $g\ kg^{-1}\ d.m.^{-1}$ | n.d. | 390 ± 19 | 330 ± 31 | 580 ± 31 | 360 ± 51 |
| TC | $g\ kg^{-1}\ d.m.^{-1}$ | 347 ± 72 | 407 ± 60 | 293 ± 55 | 326 ± 43 | 364 ± 17 |
| TN | $g\ kg^{-1}\ d.m.^{-1}$ | 150 ± 14 | 229 ± 17 * | 155 ± 33 | 141 ± 25 | 213 ± 15 |
| P | $g\ kg^{-1}\ d.m.^{-1}$ | 8.65 | 3 | 4 | 4.05 | 3.98 |
| K | $g\ kg^{-1}\ d.m.^{-1}$ | 80.8 | >100 | >100 | >100 | >100 |
| S | $g\ kg^{-1}\ d.m.^{-1}$ | 11 | n.d. | 18.8 | 5.3 | 13.4 |
| Ca | $mg\ kg^{-1}\ d.m.^{-1}$ | 17,800 | 13,684 | 23,700 | 8500 | 24,800 |
| Mg | $mg\ kg^{-1}\ d.m.^{-1}$ | 7090 | 11,670 | 11,290 | 10,210 | 12,410 |
| Na | $mg\ kg^{-1}\ d.m.^{-1}$ | >55,000 | >55,000 | >55,000 | >55,000 | >55,000 |
| Fe | $mg\ kg^{-1}\ d.m.^{-1}$ | 2710 | 216 | 400 | 530 | 800 |
| B | $mg\ kg^{-1}\ d.m.^{-1}$ | 62 | 98 | 86 | 101 | 100 |
| Mn | $mg\ kg^{-1}\ d.m.^{-1}$ | 47 | 11 | 30 | 10 | 23 |
| Zn | $mg\ kg\ d.m.^{-1}$ | 63.1 | 15.4 | 10.1 | 18.5 | 21.3 |
| Cu | $mg\ kg^{-1}\ d.m.^{-1}$ | 15.38 | <0.7 | 0.05 | 2.32 | 4.37 |
| Ni | $mg\ kg^{-1}\ d.m.^{-1}$ | 12.4 | 10.9 | 8.6 | 18.4 | 8.1 |
| Cr | $mg\ kg^{-1}\ d.m.^{-1}$ | 4.7 | 1.72 | 2 | 2.7 | 4.6 |
| Hg | $mg\ kg^{-1}\ d.m.^{-1}$ | 0.041 | n.d. | 0.005 | 0.029 | 0.014 |
| Pb | $mg\ kg^{-1}\ d.m.^{-1}$ | 3.42 | <3.5 | 0.19 | 0.63 | 1.18 |
| Cd | $mg\ kg^{-1}\ d.m.^{-1}$ | 0.1 | <0.7 | 0.02 | <0.01 | 0.01 |
| C/N | ratio | 2.3 | 1.8 | 1.9 | 2.3 | 1.7 |
| N/P | ratio | 17 | 79 | 39 | 35 | 54 |
| TAN/TN | ratio | 93% | 250% ** | 77% | 11% | 32% |

Average value of 3–6 samples/batch (sampling every two weeks) with standard deviation is shown for parameters pH, EC, $NO_3$-N, $NH_4$-N, TS, OM, TC, and TN. Dry samples from each batch were collected in one uniform sample and analyzed for other elements. n.d.—no data. * and ** comment: in D2 almost all nitrogen was in $NH_4$-N form, almost half of it seems to have been lost before analysis on CN analyzer (samples should have been acidified), which is why the value of TN, and consequently, TAN/TN, is unreal.

### 2.4. Climate and Pond Conditions

The climate data were taken from ARSO [32]. The pH in the ponds was maintained at 7.2 by controlled $CO_2$ injection. Temperature readings were taken between 7 and 10 a.m. to avoid the effects of strong solar radiation. The ponds were protected from freezing by controlling the air temperature in the greenhouse to above zero °C.

### 2.5. Sampling and Analysis

Samples of microalgae culture for the measurements of TS, $NO_3$-N, $NO_2$-N, $NH_4$-N, OD, DFI, and microscopic observation were taken directly from ponds twice a week, while digestate samples were taken directly from the container once a month. Before every sampling, fresh water was added to the culture in ponds until 20 cm culture depth was reached to eliminate the influence of water evaporation. If feeding was needed, it was performed after sampling to eliminate the influence of digestate addition on parameters values. Temperature, pH, and electroconductivity of medium were measured directly in ponds using systems for continuous automated measurements of these parameters in liquids (JBL Proflora pH-Control Touch with pH electrode and temperature sensor for automated control of the acidity and $CO_2$ content and Jishen CM-230 conductivity meter with conductivity electrode). Total solids (TS) were determined after drying the sample at 105 °C until the constant weight according to the APHA Standard Methods [33]. After centrifugation and filtration of the samples over 0.45 μm, $NO_3$-N, $NO_2$-N, and $NH_4$-N were

analyzed in filtered supernatant spectrophotometrically (Thermo Scientific™ Gallery™ Discrete Analyzer, Thermo Fischer Scientific, Vantaa, Finland, method ISO 14255 [34], Total Oxidized Nitrogen (TON—nitrite; nitrate as calculation with vanadium chloride reduction; nitrite: sulfanilamide coupling with N-(1-naphthyl)-ethylenediamine dihydrochloride; ammonia: glutamate dehydrogenase [35]. Optical density (OD) was measured twice a week in the range of 400 to 800 nm using a UV-6100S UV/VIS Spectrophotometer. The OD at 680 nm was chosen as the indicative parameter of algal growth [36]. Photosynthetic activity was measured by delayed fluorescence intensity (DFI) [37,38]. DFI was measured in a custom-made photon-counting luminometer [38,39] containing a red light-sensitive photomultiplier tube (Hamamatsu R1104, Hamamatsu Photonics, Hamamatsu City, Japan) with a photon counting unit (Hamamatsu C3866, Hamamatsu Photonics, Hamamatsu City, Japan). DFI was measured in counts per minute (cpm) in the interval from 1 to 60 s after 3 s illumination with 22 µmol m$^{-2}$ s$^{-1}$ PAR from 20 W halogen lamp through a short-pass filter ($\lambda$ < 600 nm). Photosynthetic culture index (PCI) was introduced as a sensitive indicator of culture condition based on the findings of the higher sensitivity of similar index PhAI by Berden Zrimec et al. [40]:

$$PCI = OD(680)/DFI \times 10^6 \tag{1}$$

Changes in the composition of the microalgae community were observed under a light microscope at 100× and 400× magnification (AmScope T690C-PL-5M, AmScope, USA). Organoleptic properties such as culture color and microalgae morphology, including flocculation, homogeneous distribution, settling ability, foaming, and odor, were continuously observed. After gravity sedimentation without flocculants, the harvested microalgal biomass and an aliquot of the liquid digestate were air-dried at 40 °C for further analyses (TN, TC, micro- and macronutrients). Total C and N content was determined in the air-dried samples by dry combustion (Vario Max CN Element Analyzer, Elementar, Analysensysteme GmbH, Hanau, Germany, ISO 10694:1995) [41]. The content of macro- and micronutrients in dry digestate samples was analyzed in a 1 g sample, split digested in HNO$_3$ and then in aqua regia, and analyzed by ICP-MS for ultra-low detection limits [42].

### 2.6. Experiment Stages

The microalgae cultivation experiment was divided into three stages (Figure 3):

- Learning/Design Stage (SI): initial monitoring from February 2019 to July 2019 to establish testing conditions. The algae were fed by untreated digestate, increasing from 2 L day$^{-1}$ in February 2019 to 27 L day$^{-1}$ in June 2019 (corresponding to from 0.42 up to 4.38 g total daily nitrogen input, TN$_{in}$ m$^{-2}$ day$^{-1}$, resulting in EC levels between 1900 µS cm$^{-1}$ in March 2019 and max. 5800 µS cm$^{-1}$ in June 2019). Inoculum OD680 at the beginning of SI was 0.57.
- Development/Testing Stage (SII): three ponds were exposed to varying digestate treatments and EC levels in July 2019. P1 had low nutrition (EC < 1500 µS cm$^{-1}$), P2 had medium nutrition, presumably the most optimal (EC 1500–2500 µS cm$^{-1}$), and P3 had high nutrition (EC > 2500 µS cm$^{-1}$). These EC levels corresponded to <1, 0.5–3.0 and >2 g total daily nitrogen input (TN$_{in}$ m$^{-2}$ day$^{-1}$) in P1, P2, and P3, respectively. Inoculum OD680 at the beginning of SII was 1.06, 1.48, and 1.21 in P1, P2, and P3, respectively. This stage concluded in late August 2019 due to potential culture collapse.
- Calibration/Verification Stage (SIII): started in September 2019 and lasted until July 2020 and included the adjustment of culturing and harvesting regimes to ensure sustainable cultivation of microalgae. The conditions were as follows: average EC 712 µS cm$^{-1}$, 938 µS cm$^{-1}$, 1316 µS cm$^{-1}$; average TN$_{in}$ m$^{-2}$ day$^{-1}$ 0.3, 0.6, and 1.0 g; and inoculum OD680 at the beginning of SIII was 1.06, 1.48, and 1.21 in P1, P2, and P3, respectively.

| Experiment timelayer | | | | | Ponds | | Cultivation mode | | | | | Dynamic measurments and observations |
|---|---|---|---|---|---|---|---|---|---|---|---|---|
| Year | Months | Season | Digestate batch | Stage | Pond number | Digestate input | Cultivation | Harvesting | Medium recycling | Biomass recycling | Feeding | |
| 2019 | Feb | Winter | DIG1 | SI | P0 | low to high | semi-continous | continuous sedimentation | YES | NO | conti-nuous | pH, T, EC, TS OD DFI<600 NH$_4$-N NO$_3$-N NO$_2$-N TC TN species composition |
| | Mar-May | Spring | | | | | | | | | | |
| | Jun-Avg | Summer | DIG2 | SII | P1 | low | | | | YES | batch | |
| | | | | | P2 | medium | | | | | | |
| | | | | | P3 | high | | | | | | |
| | Sep-Nov | Autumn | DIG3 | SIII | P1 | low | | batch sedimentation | NO | NO | | |
| 2020 | Dec-Feb | Winter | DIG4 | | P2 | medium | | | | | | |
| | Mar-May | Spring | DIG5 | | P3 | high | | | | | | |
| | Jun-Jul | Summer | | | | | | | | | | |

**Figure 3.** Experimental scheme.

## 2.7. Productivity Calculations

Biomass productivity, (BP)—biomass produced in ponds between two harvesting events, was calculated as follows (according to the modified method by Marazzi et al. [43]):

$$\text{BP [g m}^{-2}\text{ day}^{-1}] = ((\text{TS}_{t1} - \text{TS}_{t0}) - \sum \text{TS}_D + \sum \text{TS}_{Mh})/((V - \Delta V) \times A \times \Delta t) \tag{2}$$

where $\text{TS}_{t1}$ is the total solids in the pond before harvesting at t1 (g L$^{-1}$), $\text{TS}_{t0}$ is the total solids in the pond after harvesting at t0 (g L$^{-1}$), $\sum \text{TS}_D$ is the sum of total solids added with digestate between t0 and t1 (g L$^{-1}$), $\sum \text{TS}_{Mh}$ is total solids harvested as microalgae biomass at t1 (g L$^{-1}$), V is volume of medium in pond (L), $\Delta V$ is change of volume due to digestate addition and harvested biomass (L), A is pond area (m$^2$), $\Delta t$ is no. of days between two harvests, t0 and t1.

Average seasonal biomass yield (BY) per area in time and relative comparison between ponds rBY (P2 pond with, presumably, optimal conditions for growth was chosen as reference pond) were calculated as whole harvested biomass from each pond in one season:

$$\text{BY [g m}^{-2}\text{ day}^{-1}] = \sum (M_h/(A \times \Delta t))/n, \tag{3}$$

$$\text{rBY}_{Pn} = \text{BY}_{Pn}/\text{BY}_{P2} \times 100\%, \tag{4}$$

where $M_h$ is the dry biomass of microalgae harvested in one harvest, A is the pond area, $\Delta t$ is the number of days between two harvests, n is the number of harvests in one season, and Pn presents the selected pond (P1, P2, or P3).

Seasonal nitrogen use efficiency—NUE (calculated according to internal utilization efficiency of a nutrient calculation (kg yield per kg nitrogen input) [44]) and relative comparison between ponds were calculated (P2 pond was reference value):

$$\text{NUE [gM}_h\text{ gTN}_{in}^{-1}] = \sum M_h \text{ per pond in season}/\sum N^{in} \text{ per pond in season}, \tag{5}$$

$$\text{rNUE}_{Pn} = \text{NUE}_{Pn}/\text{NUE}_{P2} \times 100\% \tag{6}$$

where $M_h$ is the dry biomass of microalgae harvested in one season, $N_{in}$ is the mass of total nitrogen added to the pond with digestate in this same season, rNUE is relative NUE, and Pn refers to pond P1, P2, or P3.

*2.8. Statistical Analysis*

For statistical data analysis, the open source statistical program R version 4.0.3 [45] was used with the package R Commander, a basic graphical statistical user interface for R, Rcmdr [46]. To determine the linear relationship between each variable/parameter in the microalgae production experiment, a correlation matrix was created and Pearson's correlation coefficient was calculated for each pair of parameters. Analysis of variance (ANOVA) was performed to compare the variances over the average of the different groups, and in case of significant interactions, Tukey's test was used to analyze the differences in the average values ($p < 0.05$).

## 3. Results

*3.1. Factors Influencing Microalgae Production*

Average monthly air temperatures and sun duration in Ljubljana, Slovenia [32], in 2019–2020 were characteristic of the temperate climate in the ALS6 environmental zone [27]. Pond temperatures in an unheated greenhouse were strongly correlated with outdoor temperatures (Figure 4).

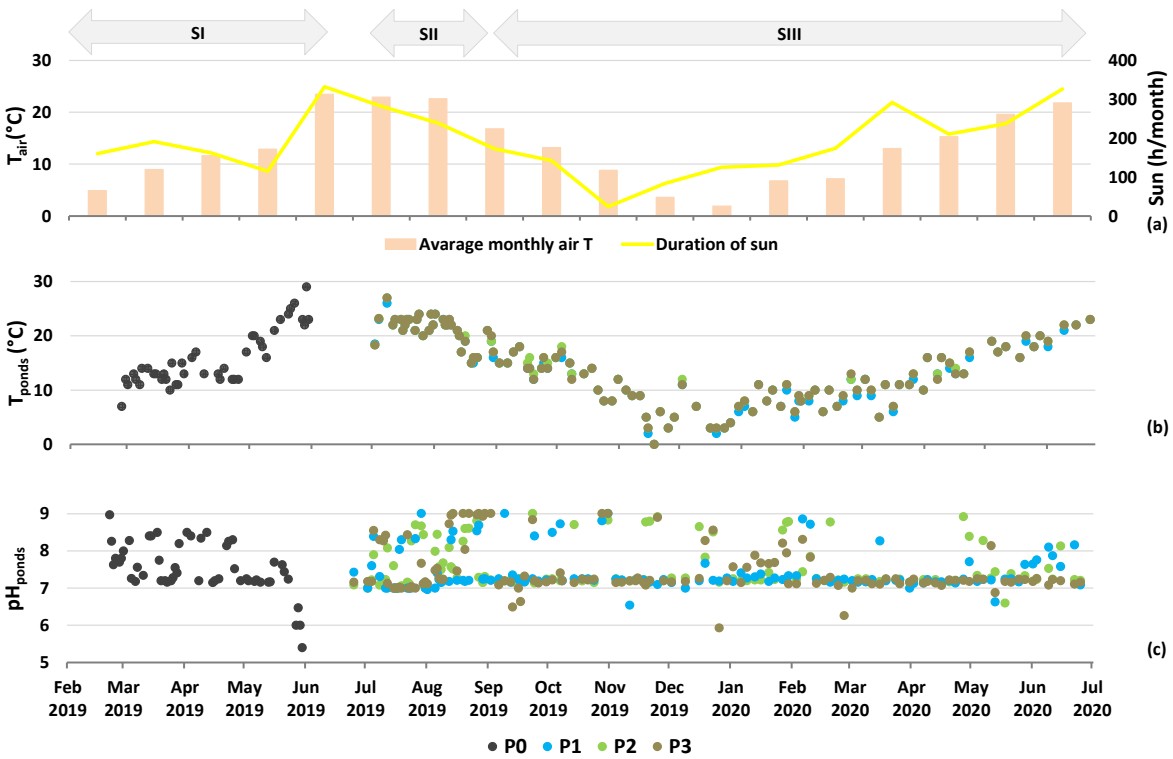

**Figure 4.** Conditions during the experiment in ponds. (**a**) Climate conditions, average monthly air temperature and sun duration in Ljubljana, Slovenia, during the experiment. (**b**) Temperature of the medium. (**c**) pH of the medium.

The nutrient composition of the five utilized digestate batches (D1–D5) differed significantly, although they were all supplied by the same biogas plant (Table 1). While batches D1, D2, and D3 were rich in mineral nitrogen (ammonium nitrogen $NH_4$-N prevailing), organic-bound nitrogen predominated in D4 and D5, as indicated by the low TAN/TN values.

The planned 1:2:3 ratio of average digestate/nitrogen feed rate (<1, 0.5–3, >2 g $TN_{in}$ $m^{-2}$ $day^{-1}$) into ponds P1:P2:P3 could be generally maintained with digestate batches D2 and D3 to maintain the planned nutrient intensity of the ponds at low–medium–high electrical conductivity. In contrast, digestate batches D4 and D5 had lower mineral N contents (Table 2), and it was not possible to simply compensate for the planned nitrogen input with higher digestate input because the systems reacted negatively in the short term; for example, algae began to flocculate or the nitrite concentration increased significantly.

**Table 2.** Input of digestate.

| Stage | Months | Nutrient Source | Average Input of Digestate (L $m^{-3}$ $day^{-1}$) | | | | Average Nitrogen Supply Rate (g $N_{in}$ $m^{-2}$ $day^{-1}$) | | | |
|---|---|---|---|---|---|---|---|---|---|---|
| | | | P0 | P1 | P2 | P3 | P0 | P1 | P2 | P3 |
| SI | March–June | D1 | 8.9 ± 5.4 | | | | 2.26 ± 1.3 | | | |
| SII | July–September | D2 | | 2.8 ± 2.1 | 8.6 ± 5.9 | 11.5 ± 7.0 | | 0.79 ± 0.52 | 1.94 ± 1.29 | 2.60 ± 1.61 |
| SIII | October–December | D3 | | 2.6 ± 1.1 | 6.1 ± 5.5 | 8.8 ± 5.2 | | 0.35 ± 0.20 | 0.68 ± 0.50 | 1.28 ± 0.94 |
| | January–May | D4 | | 1.2 ± 0.8 | 2.3 ± 1.4 | 3.9 ± 2.8 | | 0.17 ± 0.08 | 0.34 ± 0.16 | 0.56 ± 0.32 |
| | May–July | D5 | | 1.5 ± 1.0 | 3.7 ± 2.5 | 5.8 ± 4.5 | | 0.23 ± 0.16 | 0.46 ± 0.32 | 0.72 ± 0.48 |

$N_{in}$—nitrogen added to ponds with digestate. D1–D5—digestate batches. Frequency of feeding was 1.00, 0.58, 0.27, 0.22, and 0.32 $day^{-1}$ in SI March–June, SII July–September, SIII October–December, SIII January–May, and SIII May–July, respectively. Values shown in the table are the average values of all feedings in each season with standard deviation.

### 3.2. Dynamics of Physio-Chemical and Biological Parameters during the Experiment

In SI and SII stages, total solids (TS), electroconductivity (EC), and $NH_4$-N concentration in the cultures increased along with the cumulative addition of the digestate to the ponds and dropped only when the feeding was terminated. The switch to batch harvesting in SIII enabled sustaining more constant values in these parameters over the long term.

The differentiation of ponds based on the added digestate concentration was most evident in TS, EC, and $NH_4$-N values, while $NO_3$-N and $NO_2$-N concentrations were not so responsive to the digestate addition because they are more dependent on biological processes. $NH_4$-N, $NO_3$-N, and $NO_2$-N values changed significantly with the digestate quality change in the spring and summer of 2020, when D4 and D5 were fed (Figure 5).

### 3.3. Dynamics of Biological Parameters in Ponds: DFI, OD, and Species Composition

The predominant algae species in all three ponds was *Scenedesmus dimorphus*, which occurred either as single cells (mostly in spring and summer), in four-cell colonies, or in a mixture of both (fall and winter). Other species in the ponds varied slightly by pond and season but mostly consisted of *Scenedesmus quadricauda*, *Scenedemus* sp. in two-cell colonies, and *Dictyosphaerium* sp. Diatoms were present only occasionally and mostly in flocs, similar to some cyanobacteria (especially in summer 2020). Flocs of round, unidentified bacteria were observed sporadically throughout the year but occurred in greater numbers in ponds P2 and P3 during the spring and summer of 2020, when concentrations of organic matter and organically bound nitrogen were high in the ponds due to feeding with low $NH_4$-N digestates D4 and D5. *Dictyosphaerium* sp. colonies were also present in higher numbers in spring and summer 2020. Grazers were observed in limited numbers during all seasons, primarily rotifers, water fleas, and vorticella. Flocs of bacteria and algae formed at various stages but appeared in higher numbers and sizes mostly in the spring and summer of 2020 (Figure 6, Table A1).

After starting the culture in February 2019, it took one month for photosynthetic activity to increase significantly from DFI 0.5 to 2 M cpm, reaching a maximum DFI of 3.7 M cpm in SI in spring 2019. After starting the three ponds with the adapted culture (SII), photosynthetic activity was comparable in all three ponds in fall and winter 2019 (DFI 0.5–2 M cpm), while it was generally highest in P3 and lowest in P1 in spring (2020) and summer (2019, 2020). In January 2020, photosynthetic activity increased to 3.4 M, 4.3 M, and 4 M cpm in P1, P2, and P3, respectively. A similar increase was observed in August

2019, when DFI was 2.3 M and 4.4 M cpm in P2 and P3, respectively. OD increased to 2.8 in the initial stage (SI). It was kept around $1.2 \pm 0.5$ in SII and $1 \pm 0.5$ in SIII, until it increased to $1.6 \pm 0.6$ after the increase in photosynthetic activity in January 2020, with P2 having the highest OD of the three ponds (Figure 7).

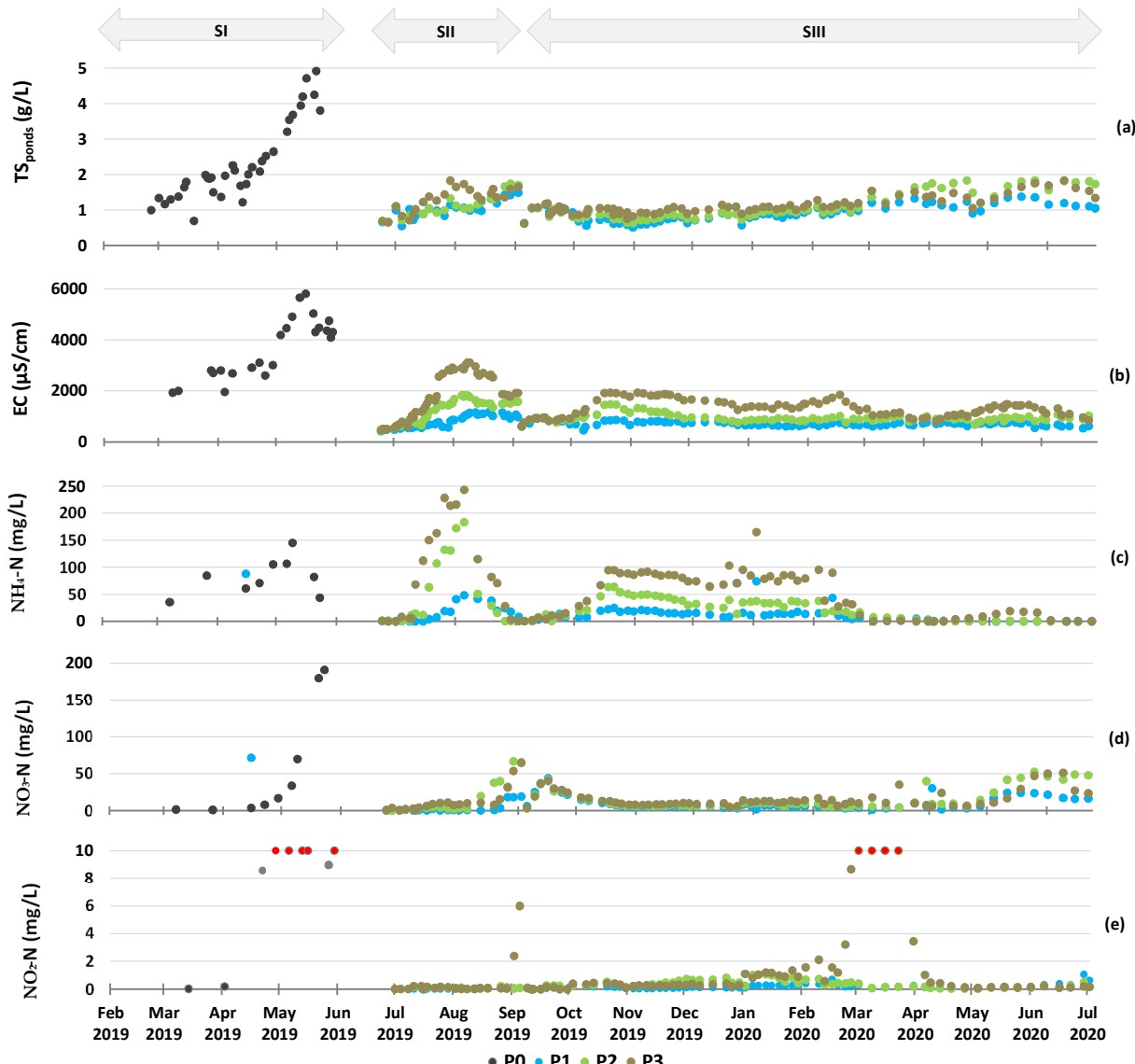

**Figure 5.** Dynamics of physio-chemical parameters measured in algal cultures in ponds P0, P1, P2, and P3 during the experiment: (**a**) total solids, (**b**) electroconductivity, (**c**) ammonium nitrogen, (**d**) nitrate nitrogen, and (**e**) nitrite nitrogen. In the case of $NO_2$-N, values above 10 mg $L^{-1}$ are shown as red dots (actual values were 13.4, 27.2, 37.5, 35.3, and 11.9 mg $L^{-1}$ in P0 SI; 14.0, 27.4, 10.6, 24.0 mg $L^{-1}$ in P3 SIII).

Photosynthetic activity declined several days before the visible changes in the cultures, indicating the need for remedial action. However, when the algae flocculated into visible flocks, photosynthetic activity decreased, but the algae remained vital and recovered on their own. In this case, PCI remained low, indicating that no action was required (P3 in March 2020, P1 in May 2020). On the other hand, the PCI value increased to 8.1 before the collapse of the P3 culture in 2019 (Figure 7). During the optimal growth of the cultures, the PCI value remained low and was mostly between 0.4 and 2.1. Therefore, a PCI value of

>2.5 was chosen as an indicator of deterioration in the condition of the cultures, but this needs further investigation.

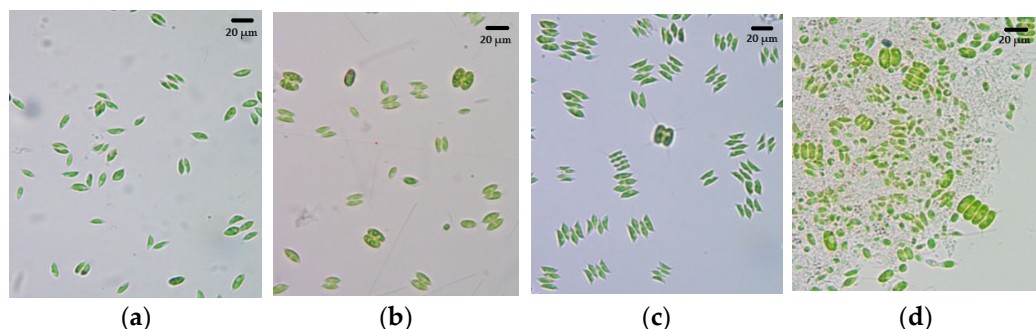

**Figure 6.** Most representative algae communities in the digestate-fed ponds under light microscope (400× magnification): (**a**) single *Scenedesmus* cells, (**b**) double-cell *Scenedesmus* colonies, (**c**) prevailing four-cell *Scenedesmus* colonies, (**d**) algae in flocks.

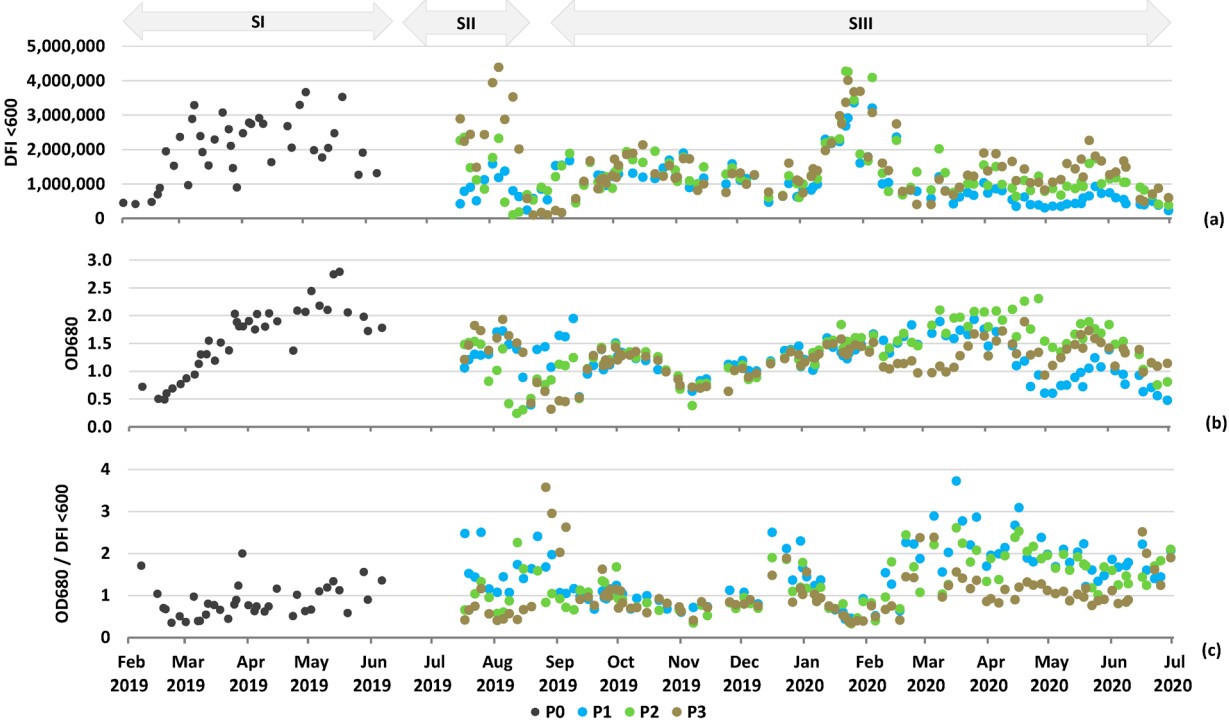

**Figure 7.** Biological parameters in ponds: (**a**) Delayed fluorescence intensity (DFI < 600), (**b**) Optical density (OD680), and (**c**) PCl—coefficient OD680/DFI < 600 during the whole study period.

*3.4. Biomass Productivity, Seasonal Biomass Yield, and Nitrogen Utilization Efficiency*

Biomass productivity BP (g m$^{-2}$ day$^{-1}$) varied among seasons and was generally highest in summer months and lower in winter (Figure 8). Differences among ponds were most apparent in the spring and summer months, when productivity was highest. In these months, pond P2 had the highest productivity, up to 10.0 g m$^{-2}$ day$^{-1}$ in SII summer. The productivity of pond P3 was always lower than that of pond P2, while pond P1 had lower productivity in the summer and spring months and the highest productivity among all three ponds in fall and winter.

Similarly, biomass yield varied between seasons, and the amount of biomass harvested was reflected not only because of seasons, the quantity, and the quality of the digestate but also as a result of harvesting efficiency (i.e., sedimentation). The biomass yield (g $M_h$ m$^{-2}$ day$^{-1}$) was generally higher in the summer months. When biomass yields were compared between ponds, more biomass was harvested in P2 and P3 than in P1, but the difference between P2 and P3 was not significant (Table 3). Thus, the higher concentration of digestate in P3 did not result in a higher biomass yield.

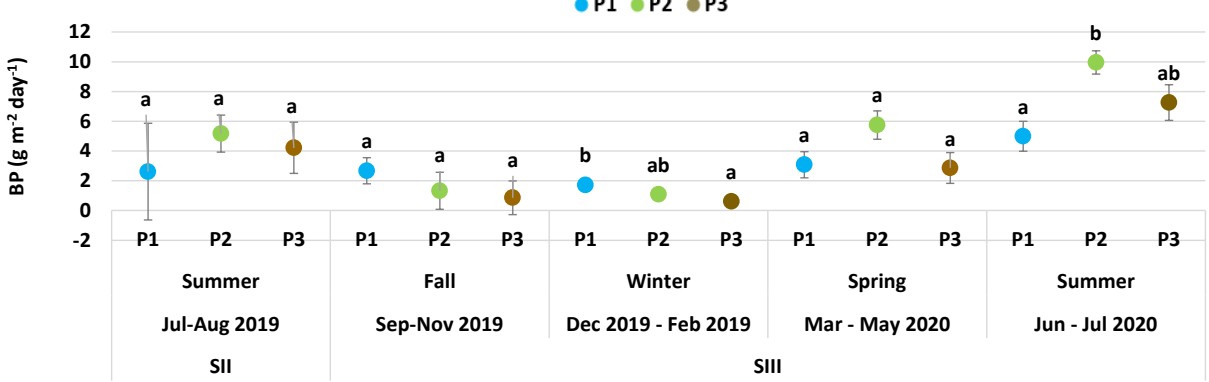

**Figure 8.** Average seasonal biomass productivity in stages SII and SIII. Average values with standard errors for biomass productivity BP (g m$^{-2}$ day$^{-1}$) in each season are presented (number of periods between two harvests: n = 22, 11, 5, 9, and 6 in SII summer, SIII fall, winter, spring, and summer, respectively). Values with different letters are significantly different at $p < 0.05$.

**Table 3.** Seasonal biomass yield and nitrogen utilization efficiency in stages SII and SIII.

| Stage | Season | Average BY (gM$_h$ m$^{-2}$ day$^{-1}$) | | | Relative BY (P2 = 100%) | | | NUE (gM$_h$ gN$_{in}$$^{-1}$) | | | Relative NUE (P2 = 100%) | | |
|---|---|---|---|---|---|---|---|---|---|---|---|---|---|
| | | P1 | P2 | P3 | P1 | P2 | P3 | P1 | P2 | P3 | P1 | P2 | P3 |
| SII | summer | 4.8 a | 13.1 b | 10.5 ab | 38% | 100% | 86% | 11.4 a | 6.9 a | 4.7 a | 164% | 100% | 68% |
| SIII | fall | 9.1 a | 9.9 a | 10.9 a | 91% | 100% | 112% | 22.3 a | 13.9 a | 11.0 a | 160% | 100% | 79% |
| | winter | 2.6 a | 2.7 a | 3.1 a | 96% | 100% | 117% | 28.5 a | 15.4 a | 11.6 a | 185% | 100% | 76% |
| | spring | 7.0 a | 9.8 a | 8.3 a | 72% | 100% | 85% | 48.1 b | 27.5 ab | 16.8 a | 175% | 100% | 61% |
| | summer | 7.8 a | 11.1 b | 10.4 b | 70% | 100% | 94% | 44.8 a | 28.9 a | 24.9 a | 155% | 100% | 86% |
| | Average ± stdev | 6.1 ± 2.6 | 9.0 ± 3.9 | 8.6 ± 3.4 | 73% ± 23% | 100% | 99% ± 15% | 31.0 ± 6.9 | 18.5 ± 4.2 | 13.8 ± 3.4 | 168% ± 5% | 100% | 74% ± 4% |

BY—biomass yield, M$_h$—dry biomass of harvested microalgae in season, NUE—nitrogen utilization efficiency, N$_{in}$—nitrogen input in season. BY values are presented as average values of n = 27 for SII summer, n = 12 for SIII fall, n = 6 for SIII winter, n = 8 for SIII spring, and n = 7 for SIII summer (n—number of biomass harvests). NUE values are presented as average values of n = 6 for summer SII, n= 6 for fall SIII, n = 5 for winter SIII, n = 9 for spring SIII, and n = 5 for summer SIII (n—number of periods in season). Values with different letters are significantly different at $p < 0.05$.

Biomass yield per unit of N input, BY (M$_h$ N$_{in}$$^{-1}$), was highest in pond P1. Pond P2 was the pond with the highest biomass yield and also expressed relatively good nitrogen utilization efficiency. Supplying pond P3 with the highest nitrogen input resulted in a 26% lower average nitrogen utilization efficiency compared to pond P2, while biomass yield was similar (Table 3). The low nitrogen supply in P1 resulted in an average 68% higher nitrogen utilization efficiency compared to P2; however, the yield was 27% lower on average (Table 3).

### 3.5. Decision Support Tool Development

The DST was developed during the 18-month operation of raceway ponds, focusing on easy-to-measure parameters like pH, T, EC, TS, $NH_4$-N, $NO_3$-N, $NO_2$-N, DFI, and OD. In the initial learning stage (SI), optimal performance was achieved, with approximately 10–15 L of digestate per pond daily (2.17–2.92 g $TN_{in}$ $m^{-2}$ $day^{-1}$). Optimal reference values were set at 2000 $\mu S$ $cm^{-1}$ for EC, 100 mg $L^{-1}$ for $NH_4$-N, less than 20 mg $L^{-1}$ for $NO_3$-N, and nitrite ($NO_2$-N) below 3 mg $L^{-1}$ (Table 4). Critical points such as maximum operational electrical conductivity and ammonium–nitrogen levels were identified, and it was recommended to switch from continuous feeding to batch feeding to improve control over the system.

**Table 4.** Reference values for setting stage SII according to SI values.

| Stage | Pond | pH | Average Amount of Digestate (L $pond^{-1}$ $day^{-1}$) | $TN_{in}$ (g $m^{-2}$ $day^{-1}$) | EC in Pond ($\mu S$ $cm^{-1}$) | $NH_4$-N in Pond (mg $L^{-1}$) | $NO_3$-N in Pond (mg $L^{-1}$) | $NO_2$-N in Pond (mg $L^{-1}$) | OD680 | DFI < 600 (cpm) |
|---|---|---|---|---|---|---|---|---|---|---|
| SI | P0 | 7.20 | 10–15 | 1.6–2.4 | 2000 | 100 | <20 | <3 | 0.7–1.3 | 0.5–3.5 M |
| SII | P1 | 7.20 | 5–10 | <1 | <1500 | 25–50 | <5 | <3 | 0.7–0.9 | 0.5–1 M |
|  | P2 | 7.20 | 10–15 | 0.5–3.0 | 1500–2500 | 50–150 | <20 | <3 | 0.9–1.1 | 1–2 M |
|  | P3 | 7.20 | 15–30 | >2 | >2500 | 150–300 | <50 | <3 | 1.1–1.3 | 2–3.5 M |

Reference values in all three ponds were tested in stage SII. Despite the initial adjustment to the set points, negative system responses such as the flocculation of microalgae and culture collapse were observed, especially in P2 and P3. The study found that the model was well configured, but there were shortcomings in the harvesting system. The continuous harvesting of algae by natural sedimentation and recirculation of the supernatant probably led to the gradual intoxication of algal culture. The presence of nitrite nitrogen ($NO_2$-N) at a higher concentration (>3 mg $L^{-1}$) was identified as a sign for enhanced caution and culture performance observation. Actions like stopping digestate addition were taken to stabilize the culture.

To remedy this, the harvesting system was changed to batch harvesting in stage SIII, resulting in more stable production and fewer problems with the culture. Algae were able to process higher levels of digestate. However, challenges arose with the introduction of D4 digestate (very low mineral nitrogen content but relatively high organic matter; TAN/TN = 11%), resulting in presumably increased microbial mineralization activity and, consequently, high nitrate and nitrite levels. Mitigative actions were taken to stabilize the system, such as stopping digestate feeding and diluting the culture with fresh water.

The DST operating protocol was tested and refined during the SIII stage, emphasizing the importance of monitoring indicators of culture vitality. A table of indicator parameters was created in which values were classified as low, optimal, high, or critical for culture growth (Figure 9). The DST was presented as a straightforward tool for maintaining algae cultures fed with digestate. Key actions during critical events included halting digestate feeding, harvesting and diluting the culture, and inoculating the culture to fresh medium if other measures failed (Figure 9).

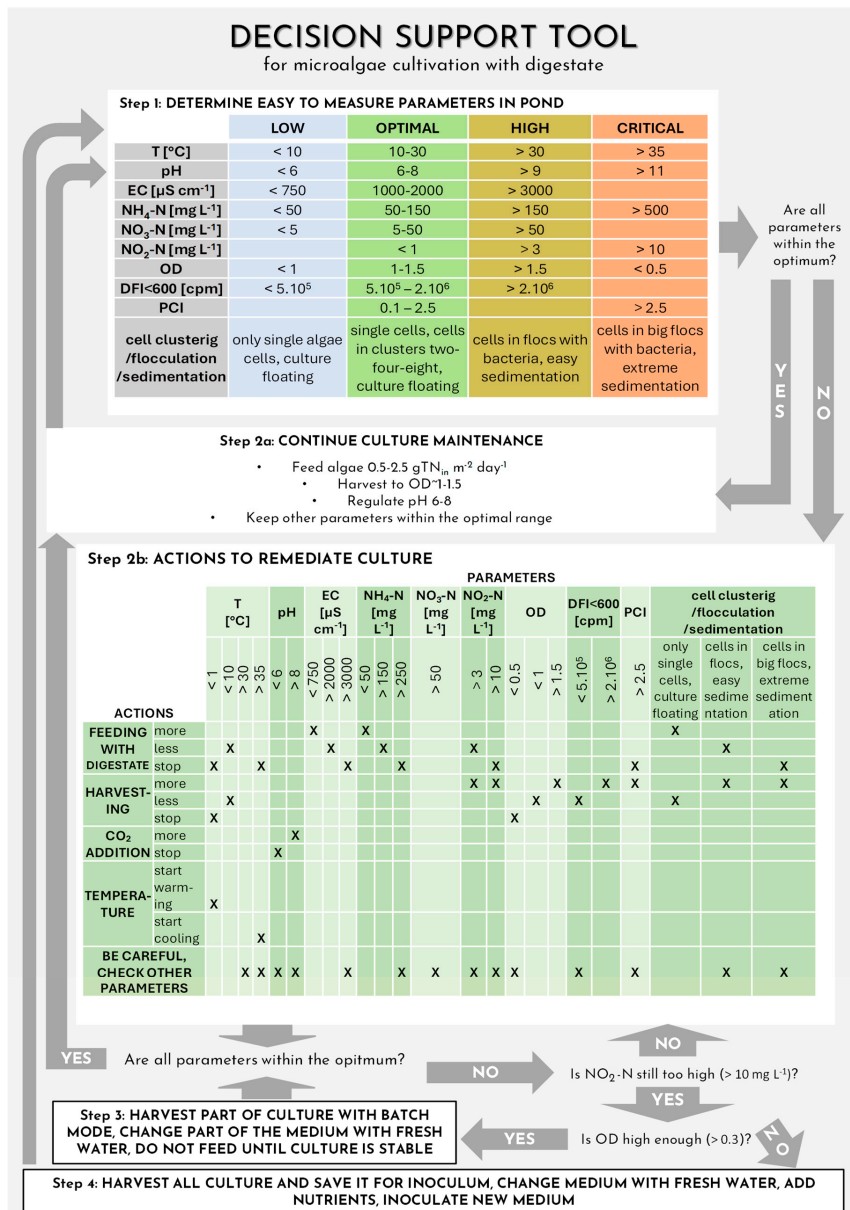

**Figure 9.** Decision support tool for microalgae cultivation on digestate. Only a few easy-to-measure parameters enable decisions about actions to be taken for stable culture maintenance and steps to save culture at critical points.

## 4. Discussion

In the sub-alpine region of Slovenia, in Ljubljana, a study was conducted over a period of one and a half years to investigate the cultivation of microalgae, primarily of the genus *Scenedesmus*, on food waste biogas digestate. Using easily measurable parameters and observations, we developed a simple decision support tool (DST) to enable sustainable microalgae production in low-cost open ponds, even under less optimal temperate conditions. The growth and production of the microalgae culture were primarily influenced by two variables: changing light availability and temperature throughout the four seasons of a year, and the quantity and quality of digestate introduced into the ponds. In addition to determining the optimal parameter values to maximize microalgal productivity, measures to intervene at critical points were proposed and validated during the three stages of the experiment.

Most European countries (including Slovenia), have sub-optimal climatic conditions for large-scale outdoor microalgae production [47], the greatest bottleneck for microal-

gae cultivation being light availability due to latitude and low solar radiation and short day lengths in colder seasons [48], limiting photosynthetic efficiency [49]. With the average yields of algae biomass in our experiment (min. 2.6 g m$^{-2}$ day$^{-1}$ in P1 in winter, maximum 13.1 g m$^{-2}$ day$^{-1}$ in P2 in summer, and annual average of 9.0 g m$^{-2}$ day$^{-1}$ (39.0 t ha$^{-1}$ year$^{-1}$) in our best-performing pond P2 (Table 3), our results are consistent with studies conducted elsewhere in Europe, such as in northern Italy [50] and the Netherlands [48].

In addition to climate, the amount and quality of digestate had a significant impact on biomass yield. When a low amount of digestate was used (P1), the biomass yield was generally lower. We only compared three concentration levels of digestate, so it is difficult to say what the optimal feed rate would be, but increasing the amount of digestate to levels above 1500 μS cm$^{-1}$ EC or a daily total N feed of more than 2.5 g m$^{-2}$ day$^{-1}$ did not result in higher yields (Table 3, Figure 8); instead, it often caused problems with the culture. The quality of the digestate varied greatly throughout the year (Table 1), which was a major challenge for the long-term microalgae production experiment. The composition of the liquid digestate can vary due to factors such as feedstock characteristics, microbial community, AD process control, and AD digester configuration [51]. The data showed that all digestate batches had lower C/N ratios (between 1.7 to 2.3) and higher N/P ratios (between 17 and 79) than the ideal ratios reported for microalgae production (C/N ratio of 4–8 and N/P ratio of 7) [51]. Soluble phosphorus levels were always low and could limit productivity in P2 and P3, while in the pond with low digestate input in P1, nitrogen was probably limiting as it was used very efficiently (44.8 g dry algae biomass produced per each g of N input). NUE at higher digestate loading decreased to 28.9 and 24.9 g biomass/g N in P2 and P3, respectively, which is still high efficiency, compared to other studies; e.g., 7 to 20 g biomass per g N input microalgae grew in different concentrations of swine manure effluent [52] and 10–20 g biomass per g N input in aquaculture effluent [53]. In our biogas digestate pretreatment for microalgae cultivation, phosphorus was predominantly retained in the solid fraction after separation [54], resulting in a digestate containing between 3.00 g kg$^{-1}$ d.m.$^{-1}$ (N/P ratio 79) in D2 to max. 8.65 g kg$^{-1}$ d.m.$^{-1}$ (N/P ratio 17) in D1 (Table 1). Furthermore, although the dilution of digestate is an efficient strategy to mitigate the inhibitory effects of free ammonia nitrogen [55], it can pose challenges in microalgae cultivation by creating nutrient deficiencies, particularly in phosphorus [56]. While the optimal phosphorus concentration for the hydroponic growth of horticultural plants falls between 30 and 60 mg P/L [57], successful *Scenedesmus* growth requires significantly lower phosphorus concentrations. Research by Ji et al. [58] demonstrated that *S. obliquus* achieved 83% phosphate uptake at an initial concentration of 4.3 mg L$^{-1}$ in piggery wastewater, highlighting the adequacy of lower phosphorus concentrations for algal growth. The data indicate that microalgae are capable of producing high amounts of biomass even under nutrient-limited conditions, making them very efficient in terms of nutrient recycling.

The liquid digestate is rich in micronutrients, crucial for optimal microalgae performance [18]. Elements such as iron and magnesium play vital roles in photosynthesis, favoring $CO_2$ fixation and biomass production, while zinc, copper, and manganese are associated with cell division [59]. Although our research did not extensively explore the impact of microelements on microalgae performance, variations in micronutrient content among digestate batches, particularly in magnesium, iron, manganese, copper, and nickel (Table 1), were evident. These differences may potentially affect microalgae growth efficiency and pose toxicity risks [18], underscoring the need for future investigations into this aspect.

To enhance productivity and nutrient utilization efficiency, regulating nutrient content in digestate, especially in the C/N and N/P ratios, is a viable consideration. Con-trolling the N/P ratio has shown benefits in biomass production, algal community stability, and effective nutrient uptake in water treatment [60]. Various options exist for improving phosphorus availability and the N/P ratio in liquid digestate as a growth medium for microalgae. While inorganic phosphorus addition (e.g., $K_2HPO_4$ or $Na_5P_3O_{10}$) is effective [61], it may not be the most convenient from a circular economy perspective. An alternative

involves using struvite precipitation for liquid digestate pretreatment, resulting in a supernatant with good properties for microalgal growth regarding $NH_4^+$-N concentration, salinity, and the N:P ratio [62]. The addition of phosphoric acid ($H_3PO_4$) serves as another P-source, simultaneously regulating microalgae medium pH, crucial for maintaining P solubility. Phosphate availability can be reduced at higher a pH, so keeping the pH close to neutral or even slightly acidic improves the P solubility [63]. Algae culture has a tendence to constantly raise the pH of a solution due to $CO_2$ consumption and $O_2$ production [64]. In our system, we regulated pH by automatically adding $CO_2$ to a pond, employing inorganic $CO_2$ addition simultaneously as a strategy to enhance the non-optimal C/N ratio of the liquid digestate. Several options were described to increase the C/N ratio when cultivating microalgae on liquid digestate. Biasiolo et al. used [55] the ultrasonic cavitation (UC) process combined with carbon dioxide ($CO_2$) insufflation for faster solubilization of the $CO_2$ in the medium and an increase in the C/N ratio of digestate. Another approach to overcome carbon deficiency could be supplementing the medium with the organic source (e.g., glucose, acetate) along with the digestate and promoting a mixotrophic mode of cultivation [65]. But, this approach can have two disadvantages: increasing the total cost of the process [65] and potential competition with heterotrophic bacteria in mixed microalgae–bacteria communities [17].

The DST includes parameters that proved to be indicative of the changes in cultures that require mitigation to preserve production. The optimal parameter values in the DST were determined based on findings from the year-and-a-half experiment as well as the existing literature and experience from previous studies on microalgae cultivation in wastewater [66]. As depicted in Figure 9, this protocol outlines the steps to achieve the most favorable conditions for microalgae production. If all measured parameters fall within the green optimal range, cultivation should continue with regular feeding, harvesting, and pH control to maintain these conditions. However, if any parameter deviates from the optimal range, specific actions may be required to restore the culture's status. These actions could include adjusting the digestate input, optimizing the intensity of harvesting and/or culture dilution, finetuning the pH by adding $CO_2$, and ensuring temperature control. In the rare cases where the culture faces imminent collapse despite these measures, the introduction of the rest of the culture into the fresh medium is considered as a last-resort solution.

Temperature is a crucial environmental factor influencing microalgae growth, biochemical composition, cell size, and nutrient consumption [67]. Microalgae can grow optimally in a wide range, between 15 and 40 °C, with lower temperatures slowing growth due to photoinhibition and slower nutrient uptake, while temperatures above the optimum sharply decrease and stop the growth [59,68]. It is important to note that the optimum temperature for microalgal growth depends on the species and environmental conditions. For instance, *Scenedesmus* sp. was successfully grown between 10 and 30 °C [67]; the optimal temperature for *Scenedesmus obtusiusculus* under normal growth conditions and nitrogen availability was 35 °C, while temperatures above 35 °C provoked the sharp decrease in photosynthetic activity, and the optimal temperature was lower in nitrogen-depleted conditions (28.5 °C) under the same irradiance conditions [69]. Our experimental results are consistent with these findings, showing increased growth rates during warmer seasons, stress-induced growth inhibition above 35 °C in summer (based on visual observations), and reduced biomass yield during winter months (Figure 8). In DST, the optimal T range for growing *Scenedesmus* mixed culture on digestate was set at 10–30 °C.

The optimal pH range in the DST was set between 6 and 8, and pH regulation (to 7.2) was achieved early in the experiment by adding $CO_2$. This decision was based on two important considerations: (i) when microalgae are supplied with digestate, the primary nitrogen source is $NH_4^+$, which converts to toxic $NH_3$ at an elevated pH [70], and (ii) the optimal pH range for the growth of microalgae belonging to genus *Scenedesmus* falls within the neutral range, typically above 6 and below 9 [69,71]. At an optimal pH, the photosynthetic activity of microalgae increases, and during the day, the pH also gradually increases due to the uptake of inorganic carbon, which limits $CO_2$ availability and, thus,

inhibits cell growth [72,73]. Simultaneously, once the medium's pH exceeds a certain threshold, the toxicity of free ammonia may hinder the growth of microalgae [74,75]. Therefore, strategies to control pH are particularly important in algal systems where ammonia serves as the predominant nitrogen source [70].

Electrical conductivity (EC), which represents the total salt concentration and increases with the addition of ions (nutrients), plays an important role in hydroponic production, with an optimal range typically between 1000 and 4000 $\mu S\,cm^{-1}$, depending on the cultivated plant species [76,77]. Although EC is not generally considered a critical parameter to measure in microalgal cultivation [66,78], and limited information exists in the literature on optimal EC values or its use in nutrient management, it has been included in DST due to its ease and speed of measurement. EC proved its value as a fast decision indicator for microalgae nutrition in the SI phase when continuous harvesting was performed, and it also showed a strong correlation with the quantity of digestate input (R = 0.94), $NH_4$-N concentration (R = 0.78), $NO_3$-N concentration (R = 0.91), and total solids (R = 0.99) in pond P0. However, it is important to note that EC is highly influenced by digestate quality and should be used for informational purposes only. Consequently, a parallel measurement of nutrient content (e.g., $NH_4^+$, $NO_3^-$, $PO_4^{3-}$ ions) with other methods is essential to ensure accurate nutrient management. In our case, most of the EC was likely attributed to $Na^+$ ions based on digestate analysis results (Table 1). In a study by Zafar et al. [79], it was also found that $CO_3^{2-}/HCO_3^-$ ions contribute to EC estimation in microalgal cultures, and salinity based on EC does not necessarily correspond to chloride concentration, leading to the conclusion that EC should not be used as the sole measure of salinity removal by algae. In the DST, the optimal range for electrical conductivity (EC) was set between 750 and 1500 $\mu S\,cm^{-1}$. However, it is important to emphasize that the medium's EC is significantly influenced by the quality of the digestate, with the addition of D1, D2, and D3 resulting in higher EC compared to D4 and D5. Consequently, EC should be used throughout the growing process based on empirical experience, and the optimal range should always be adjusted to the quality of the digestate and the results of parallel measurements of other parameters (such as nutrients, salinity, and total solids). It is known from the literature that the microalgae of the genus *Scenedesmus* are halotolerant, as are many other species commonly grown on digestate. For example, Figler et al. [80] conducted experiments cultivating microalgae in different NaCl concentrations (from 500 to 20,000 mg $L^{-1}$, corresponding to 800–36,000 $\mu S\,cm^{-1}$) and successfully grew *Scenedesmus obliquus* and *S. obtusus* even at very high $Na^+$ concentrations. The growth inhibition of microalgae when using high EC substrates rich in ammonia–nitrogen (e.g., biogas digestate) is primarily due to the toxic effects of ammonia or other inhibitory substances rather than the salts themselves [81,82], so the dilution of these feedstocks is necessary to reduce the inhibitory effect, which also reduces EC [83].

The nitrogen removal capacity by assimilation with the microalgae biomass is limited by the maximal productivity in the system, which is determined by the available solar radiation [83,84]. In temperate climates, a maximum of 3.5 g N $m^{-2}\,day^{-1}$ could have been removed if the maximum mean productivity of 50 g $m^{-2}\,day^{-1}$ had been achieved [84]. In low-production open ponds fed by digestate, the nitrogen supply is lower, especially when light availability is limited. The DST set optimal values between 0.5 and 2.5 g $TN_{in}\,m^{-2}\,day^{-1}$ (Figure 9). However, these values for $TN_{in}$ could not be achieved at all stages of the experiment. For example, the actual $TN_{in}$ average in our best-performing pond P2 with digestate D4 in spring 2020 and D5 in summer 2020 was only 0.34 g TN $m^{-2}\,day^{-1}$ and 0.46 g TN $m^{-2}\,day^{-1}$, respectively. These values were significantly lower than planned, and the addition of more digestate resulted in short-term problems such as nitrite occurrence and flocculation. This highlights that digestate input should not depend on nitrogen supply rates or nutrient measurements only and that, similar to decisions made based on EC value, close monitoring of culture vitality should also be considered.

The highest nitrogen utilization efficiency (NUE), expressed as biomass yield per unit of nitrogen input, was observed in pond P1 with the lowest nitrogen input, while in P3, NUE decreased with a high nitrogen input (Table 3). In general, limited nitrogen in the culture medium tends to decrease biomass production due to reduced photosynthetic activity caused by a decrease in active proteins in the PSII center's reaction [69,85]. On the other hand, despite reduced biomass yield, nitrogen starvation conditions may enhance microalgae photosynthetic efficiency [69] and improve the nitrogen yield coefficient [86].

The digestate quantity and composition significantly affected mineral nitrogen concentrations ($NH_4$-N, $NO_3$-N, and $NO_2$-N) in the ponds. Digestate batches rich in mineral N (D1, D2, D3) resulted in elevated $NH_4$-N levels but low $NO_3$-N. The opposite was observed at digestate with high organic N content (D4, D5), which resulted in lower $NH_4$-N concentrations and increased $NO_3$-N, especially during warmer months (Figure 5), possibly due to complex metabolic interactions between microalgae and bacteria. Occasional $NO_2$-N increases in ponds with high digestate inputs (P0 SI, P3 SII, and P3 SIII) coincided with $NO_3$-N increases, flocculation, sedimentation, and decreased DFI, indicating potential problems that require careful monitoring.

Although nitrites can present a nitrogen source for microalgae growth [87], higher concentrations have been shown to inhibit microalgae performance [88,89]. Although $NO_2$-N can be oxidized by nitrate oxidizing bacteria (NOB), several studies reported that $NO_2$-N can accumulate in microalgal–bacteria consortia, with adverse effects on microalgae varying by species; e.g., the concentrations above 25 mg $NO_2$-N $L^{-1}$ negatively impact *Scenedesmus* culture net oxygen production rate by hindering the electron transport chain between photosystems II and I (PS II and PS I) [88], and a $NO_2$-N concentration of 5 to 20 mg $NO_2$-N $L^{-1}$ inhibits the metabolism of *Chlorella* microalgae in terms of biomass productivity and nitrogen recovery rates [89]. Factors contributing to $NO_2$-N accumulation in microalgal culture include partial nitrification, influenced by various factors such as oxygen levels, pH, free ammonia (FA), free nitrous acid (FNA), total organic carbon (TOC) load, temperature, and retention time [90,91]. $NO_2$-N accumulation in microalgal culture tends to occur in seasons with high temperatures and excessive solar radiation, affecting the balance between ammonium oxidizing bacteria (AOB), nitrate oxidizing bacteria (NOB), and microalgae growth [92,93]. Dilution effectively promoted the competitiveness of microalgae over nitrifying bacteria and reduced $NO_2$-N accumulation [89]. Reducing the microalgal concentration and increasing the light availability to culture by higher dilution rates can stimulate microalgal growth [94]. In DST, the nitrite–nitrogen threshold for increased attention was conservatively set at 3 mg $NO_2$-N $L^{-1}$, with 10 mg $L^{-1}$ considered a critical point requiring immediate action to prevent culture problems. Further research is needed to refine these values, as dilution strategies, while effective, are not always the most practical or rational solution.

Photosynthetic activity is a sensitive indicator of changes in microalgae, whether measured with delayed or prompt fluorescence [95,96]. In our case, DFI could be used as an early-warning indicator, together with the photosynthetic culture index (PCI), a coefficient of OD (680) and DFI that fine-tuned the decision support tool. Since instruments measuring delayed fluorescence are not readily available, the DFI can be replaced by other measurements of photosynthetic activity, such as the maximum quantum yield of Photosystem II, represented by Fv/Fm. Since these parameters are measured only in living algal cells and reflect their health, combining them with OD provides additional information about the culture. For example, when the culture was still visually unchanged and biomass yield was stable but DFI dropped, PCI increased due to high OD. On the other hand, when algae were not deteriorating but flocculated, both OD and DFI dropped, while PCI remained low.

Another factor contributing to the deterioration of microalgae could be the higher organic load from digestate (Table 1), which can create conditions that favor the growth of heterotrophic bacteria, potentially leading to their dominance over microalgae. For example, in the case of P3, an increased abundance of flocs was observed compared to

other ponds when D4 was used as a feed source. Bacteria, cyanobacteria, whole cells of *S. dimorphus*, and grazers (rotifera) were also found in this pond during light microscopy examination (Table A1). Praveen et al. [97] reported an increase in absorbance, a decrease in COD, and a loss of green coloration, which could be attributed to the invasion of fast-growing heterotrophic microorganisms. This invasion is likely a result of the high organic content of the digestate and the presence of non-sterile aerobic conditions. Heterotrophic microorganisms can pre-dominate microalgae in terms of growth rates; they may starve the microalgae, and sometimes, nutrient-rich microalgae may be metabolized as well [97].

Finally, recycling the reused medium and/or some of the harvested biomass back into the microalgae culture can be a source of several challenges. One of the major bottlenecks faced by the microalgae industry in further biomass processing is the high cost of harvesting the microalgae biomass [98]. Processes that aim to produce low-cost products from microalgae (such as cultivation in open ponds, feeding with digestate or wastewater for the production of fertilizers and biostimulants) often cannot justify the costs associated with high-tech and expensive harvesting methods; in such cases, biomass sedimentation becomes a viable and cost-effective choice. The reuse of microalgae cultivation water after biomass harvesting is important for cost-saving and ecological aspects since it reduces water pumping and the costs of treatment, water, and nutrients [99]. At the same time, the recycling of a part of harvested biomass is known to improve sedimentation and, thus, harvesting efficiency [100]. However, with the recycling of the medium during the SI stage, a portion of the harvested biomass not only led to an accumulation of TS in the ponds and an increase in EC (Figure 5) but also likely resulted in the buildup of unidentified substances in our experiment, which in turn led to problems within the culture, including flocculation and increased sedimentation. For this reason, we initially chose not to reintroduce the harvested biomass in the SII stage. In the subsequent SIII stage, we switched to a batch harvesting procedure without medium recycling to proactively prevent the reoccurrence of such problems in the future. These strategic decisions not only stabilized the production rate and ensured the long term sustainability of the system but also led to an improvement in the efficiency of digestate utilization (Figure 8, Table 3).

The reuse of water in microalgae cultivation can lead to various issues connected to the accumulation of growth inhibitors, such as free fatty acids, polysaccharides, polyunsaturated aldehydes, and humic acid, negatively affecting algal growth and productivity [101]. For instance, soluble algae products, like organic bases and hydrophilic acids, inhibited the growth and lipid accumulation in *Scenedesmus* sp. LX1 [99], and enhanced osmolarity caused by the accumulation of $Cl^-$ ions and humic acid inhibited *Euglena gracilis* growth and photosynthetic efficiency [102]. Lu et al. [103] identified inhibitory mechanisms related to water reuse, such as cell debris, bacterial competition, dissolved organic matter, harvesting additives, and increased salinity, impacting microalgae growth through cell aggregation, competition, and cell damage.

The DST was designed to streamline decisions in microalgae cultivation, focusing on feeding, harvesting, and interventions to maintain the optimal conditions and prevent critical situations. Given the diverse quality of digestate, influenced by the anaerobic digestion process, growers often face the challenge of determining when and how much to feed the algae. Based on insights from our year-and-a-half-long experiment and measured parameters, we recommended additional feeding with the digestate to the microalgae community (predominated by genus *Scenedesmus*) in specific scenarios, namely, when EC drops below 750 $\mu S\, cm^{-1}$, when $NH_4$-N concentration falls below 50 mg $L^{-1}$, and when the culture exhibits only individual *Scenedesmus* cells, freely floating. Regular occurrences of these signs in pond P1, where nutrients were usually lacking, prompted the need for additional feeding. While feeding should maintain optimal conditions (EC 1000–2000 $\mu S\, cm^{-1}$, $NH_4$-N 50–150 mg $L^{-1}$, and culture still floating, but cells of *Scenedesmus* could also form clusters), it is crucial to cease feeding if parameters exceed critical values specified in the DST (Figure 9). Although our DST relies on a limited parameter set, its adaptability allows for potential expansion to a more comprehensive set, enabling sophisticated decision-

making not just on when and how much to feed but also considering the diverse qualities of digestate, a notable challenge in our experiment. The TAN/TN ratio posed a significant challenge in assessing digestate quality, hindering decisions based on TN, mineral forms of nitrogen (particularly $NH_4$-N content in the culture), and EC. In the context of microalgae cultivation on liquid digestate, dosage decisions are predominantly based on influent nutrient content (TN and TP) [18], considering maximum potential photosynthetic activity, including light availability and C-fixation potential [49]. Additionally, parameters such as turbidity, pH, color, light availability [104,105], TS, VS, and C/N ratio [106], along with COD and micronutrient availability [51], play crucial roles in determining the quantity of digestate required and the necessary modifications or pretreatment methods.

The existing DST is designed to assist in the manual management of microalgae, allowing users to make decisions on harvesting, feeding, and other actions based on parameters they define. However, with the shift towards automated systems, like Agriculture 4.0 [107], also trending in microalgae cultivation [24,26], the DST has the potential to serve as the foundation for the development of an automated system. This transition involves incorporating advanced monitoring through IoT sensor systems, accompanied by sophisticated software and a digital DST tool capable of autonomous decision-making. The automated system would oversee the dosage of nutrients and water as well as the harvesting process. However, challenges arise in achieving comparable values to laboratory analyses. While different parameters could be considered for monitoring in an automated DST system, complete independence from human intervention is hard to achieve. Factors such as microalgae vitality, the presence of grazers, and enhanced flocculation, beyond IoT control, remain critical and necessitate the expertise of an experienced grower.

## 5. Conclusions

Cultivating microalgae on digestate is associated with numerous challenges, including the initial determination of digestate suitability for cultivation and the adjustment of cultivation parameter. The development of a DST based on our experience has allowed us to create a simple tool that, if instructions are followed, allow for sustainable *Scenedesmus*-predominated microalgal consortia cultivation on digestate, even under less-than-optimal conditions. The optimal conditions identified in the experiment for *Scenedesmus*-predominated microalgae culture include temperature (10–30 °C), pH (6–8), EC (1000–2000 $\mu S\ cm^{-1}$), $NH_4$-N (50–150 mg $L^{-1}$), $NO_3$-N (5–50 mg $L^{-1}$), $NO_2$-N (<1 mg $L^{-1}$), OD680 (1–1.5), DFI < 600 ($5.10^5$–$2.10^6$ cpm), PCI (0.1–2.5), and a floating culture with *Scenedesmus* cells in various modes (single or clusters). These conditions (excluding $NH_4$-N concentration) were successfully maintained in pond P2 during summer SIII, resulting in an average algal biomass production of 11 ± 1.5 g $m^{-2}$ $day^{-1}$ and a nitrogen use efficiency of 28 ± 2.6 g biomass/g N-input. To further improve the efficiency (both economically and environmentally) of microalgae cultivation, future research should focus on improving the media (balancing the nutrients in the digestate and possible pretreatment of the digestate), improving harvesting efficiency, and investigating the potential for media recycling.

**Author Contributions:** Conceptualization, L.R., R.M. and M.B.Z.; methodology, L.R., M.B.Z., B.L., R.R. and R.M.; validation, L.R. and M.B.Z.; formal analysis, L.R. and M.B.Z.; investigation, L.R., V.Ž., B.L., A.C. and M.B.Z.; resources, B.L., R.R. and R.M.; data curation, R.M. and R.R.; writing—original draft preparation, L.R., M.B.Z. and R.M.; writing—review and editing, R.M. and R.R.; visualization, L.R., M.B.Z. and A.C.; supervision, R.M.; project administration, R.M.; funding acquisition, R.M. All authors have read and agreed to the published version of the manuscript.

**Funding:** This research was funded by Water2REturn, an Innovation Action co-funded by the European Commission under its Horizon 2020 program, Grant Agreement: 730398. The publication of results was supported by Horizon Europe project Turning food waste into sustainable soil improvers for better soil health and improved food systems, acronym Waste4Soil, Grant agreement ID: 101112708. The authors acknowledge the financial support of the Slovenian Research Agency (ARRS) within the infrastructural centers IC RRC-AG (I0-0022-0481-001 and I0-0022-0481-0481-06).

**Institutional Review Board Statement:** Not applicable.

**Informed Consent Statement:** Not applicable.

**Data Availability Statement:** The data presented in this study are available on request from the corresponding author. The data are not publicly available due to the sheer size, complexity, and proprietary nature of the datasets.

**Acknowledgments:** The authors acknowledge KOTO d.o.o., the biogas plant, for supplying the digestate.

**Conflicts of Interest:** Authors Maja Berden Zrimec, Borut Lazar, Robert Reinhardt and Ana Cerar were employed by the company Algen d.o.o. and participated as experts in the field of algae cultivation in a non-commercial study. The remaining authors declare that the research was conducted in the absence of any commercial or financial relationships that could be construed as a potential conflict of interest.

## Appendix A

**Table A1.** Dynamics in species composition in ponds during different seasons.

| | | Microalgae Species [1] and Bacteria and/or Flocs Present in the Community | | | |
|---|---|---|---|---|---|
| **Stage** | **Season** | **P0** | **P1** | **P2** | **P3** |
| **SI** | spring | *S. dimorphus*, *S. quadricauda*, *S. obliquus* separate single cells mode in case of *S. dimorphus* | / | / | / |
| | summer | *S. dimorphus*, *S. quadricauda*, *S. obliquus* some flocs, some bacteria | / | / | / |
| **SII** | summer | / | *S. dimorphus*, *S. quadricauda* mostly single cells mode, some floc seen at the end of summer | *S. dimorphus*, *S. quadricauda*, single cells mode, flocs at the end of summer | *S. dimorphus*, *S. quadricauda*, *Dictyosphaerium* sp. whole cells at the beginning and then single cells mode; other species present in higher numbers |
| **SIII** | fall | / | *S. dimorphus*, *S. quadricauda*,*diatoms*, *Dictyosphaerium* sp. dominant algae in whole cells mode, at the end flocs with bacteria seen | *S. dimorphus*, *S. quadricauda*, *cyanobacteria* dominant algae in whole cells mode, at the end flocs with bacteria seen | *S. dimorphus*, *S. quadricauda*, *cyanobacteria*, *Dictyospaherium* sp., *diatoms*, *Scenedesmus* sp. whole cells mode, flocs of bacteria seen |
| | winter | / | *S. dimorphus*, *S. quadricauda*, diatoms, *Scenedesmus* sp., cyanobacteria whole and single cells mode of *S. dimorphus*, some flocs | *S. dimorphus*, *S. quadricauda*, *Scenedesmus* sp. whole and single cells od *S. dimorphus*, some rotifera presnet | *S. dimorphus.*, *S. quadricauda*, *Scenedesmus* sp. start: single cells mode, flocs, end: single cells and whole, flocs, some cyanobacteria, and grazers present (vorticella) |

**Table A1.** *Cont.*

| | | Microalgae Species [1] and Bacteria and/or Flocs Present in the Community | | | |
|---|---|---|---|---|---|
| Stage | Season | P0 | P1 | P2 | P3 |
| SIII | spring | / | *S. dimorphus*, *S. quadricauda*, *Scenedesmus* sp. *S. dimorphus* mostly in single cell mode, at the end also whole cells; flocs, bacteria present, grazers | *S. dimorphus*, *S. quadricauda*, *Scenedesmus* sp. *S. dimorphus* in single cell mode mostly, flocs with bacteria present | ***S. dimorphus*, *S. quadricauda*, *Scenedesmus* sp., *Dictyosphaerium* sp.** more flocs than other ponds, bacteria and cyanobacteria, whole cells of *S. dimorphus*, grazers (rotifera) |
| | summer | / | ***S. dimorphus*, *S. quadricauda*, *Scenedesmus* sp.** mostly whole cell mode, flocs presnet, bacteria, grazers | ***S. dimorphus*, *S. quadricauda*, *Scenedesmus* sp.** whole cell mode, more of bacteria seen (form flocs as well), flocs present, thread-like cyanobacteria in higher numbers | ***S. dimorphus*, *S. quadricauda*, *Scenedesmus* sp., *Dictyosphaerium* sp.** bigger flocs, bacteria in flocs (round and thred-like), whole cells mode, *Dictyospaherium* sp. in big colonies; more *S. quadricauda* than *S. dimorphus* at the end of the period |

[1] S. dimorphus—Scenedesmus dimorphus, S. quadricauda—Scenedesmus quadricauda, S. obliquus—Scenedesmus obliquus. In bold—dominant species.

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
