# Peer review of "Microalgae Production on Biogas Digestate in Sub-Alpine Region of Europe—Development of Simple Management Decision Support Tool"

_sustainability, doi:10.3390/su152416948_

Round 1

Reviewer 1 Report

Comments and Suggestions for Authors

Dear Authors,

This is an interesting work, performed for quite a long time, under pilot scale conditions, using microalgae consortium and digestate that was not a complete novelty.

The manuscript is generically well written. Nevertheless, there are a few important issues that must be addressed before I can recommend this manuscript for publication:

1- In the Abstract section, Line 24, Authors stated they developed a DST tool that has "universal use for microalgae cultivation with biogas digestate".

I cannot agree with such claim, as Authors only demonstrated the possibility of using such tool with a particular consortium. Thus, Authors must rewrite such conclusion to highlight the suitability of the tool for this kind of consortium.

2- In line 43 of the Introduction section, Authors refer to the availability of fertilizers produced from macroalgae whereas this work is abaout microalgae.

The Authors should find suitable reference about fertilizer production from microalgae.

3- There are several lumped lists of references that must be avoided - Authors should not cite more than two references at a time. They, should rearrange the text placing each reference closer to the place where it is relevant, or add at least a few words/a single sentence to justify why each reference is relevant and must be cited.

4- The Materials and Methods section is missing many details to allow the replication of the work:

a) what is the area of each pond used in the experiments;

b) what is the depth of culture in the pond;

c) what is the inoculum concentration at the beginning of each assay;

d) what is the TN input per day for each assay in stage 2 and stage 3 - must provide the range corresponding to the EC level at the input;

e) the Biomass productivity accounted for the volume of sample taken, the volume of digestate added, but does not seem to have accounted for the water evaporation - what did Authors do to limit water evaporation? What was the water evaporation rate compared to the culture volume in the pond?

f) Why did Authors choose Pond 2 as the reference one?

g) methods used to monitor each parameter must be provided.

5- Results and Discussion sections:

a) digestate characteristicas are completely different in what concerns the N composition and content, as well as the C:N:P ratios. Similarly for micronutrients, which has also implications on microalgae growth. This must be highlighted and discussed further.

b) line 217, "and 0.32 day-1 " - does this mean you only added digestate once every 3 days?

c) Lines 286-292 - Does this mean that in spite of higher biomass productivity, the biomass yield was not higher?

d) Figure 8 - Biomass yield should also be added to this Figure.

e) Line 381-383 - the C/N and N/P ratios were not adequate. Then, what could you do to improve such ratios?

f) Lines 431-433 - what would be such strategies to control the pH?

g) Can the developed DST be converted to an automatic system, that is, to take the decision without the operator intervention?

h) Further discussions should be included, ex. on how to use the digestate characteristics to decide the dosage, etc.

6- The Conclusions section is too generic. Conclusions are insufficiently informative and Authors should provide numerical values to support the best results achieved, the corresponding conditions, etc., highlighting the fact that the tool was developed for a specific species consortium.

Reviewer 2 Report

Comments and Suggestions for Authors

The relevance of microalgae cultivation on degestate is obvious. The manuscript presents the results of a one-and-a-half-year study of the microalgae cultivation in anaerobic digestate from food waste. It is very valuable that in this work, the optimums for culturing microalgae on degestate were obtained and a cultivation algorithm was developed. In addition, the influence of all investigated physicochemical parameters on microalgae cultures is discussed in detail with literature data, which is an undoubted advantage of the work.

My recommendation is to extand the introduction by analysing the studies on the green microalgae efficiency in digestate treatment.

It would be good to put the author and the year at first mention of cultivated species.

Scale bars would be useful in Figure 6

Round 2

Reviewer 1 Report

Comments and Suggestions for Authors

Dear Authors,

Thank you for having taken into consideration the Reviewers' suggestions/comments.

I have only minor suggestions that can be implemented during the Proof Editing:

In the Abstract section, Line 26, I would suggest to rewrite as "DST offers the possibility to be tunned according to producers..."

In the Conclusions section, Line 692, I believe the word "allowing" should be replaced with the word "allowed".

Name of microalgae must be written in italic, first word, first letter capitalized, second word , only small letters

1- Line 644 and Line 977 - Scenedesmus;

2 - Line 825 - Nephroselmis sp., Amphidinium carterae and Phormidium sp.).

3 - Line 830 - Prorocentrum minimum

4 - Line 835 - Dunaliella tertiolecta

5- Line 881 - Pleurochrysis carterae and Emiliania huxleyi

6 - Line 883-884 and Line 916 and Line 935-936 - Chlorella vulgaris

7 - Line 899 - Xanthonema hormidioides

8 - Line 901 - Scenedesmus obtusiusculus

9 - Line 903 - Chlorella pyrenoidosa

10 - Line 941-942 - Use uniform style for this citation and write the microalga name as Nannochloropsis

11 - Line 961 - Chlorella

12 - Line 984 - Euglena gracilis

13 - Line 878 - it should be "pH"

Additionaly, in the list of references, for those that are not written in English language, please add "in Language"